# Transcription Factor SrsR (YgfI) Is a Novel Regulator for the Stress-Response Genes in Stationary Phase in *Escherichia coli* K-12

**DOI:** 10.3390/ijms23116055

**Published:** 2022-05-27

**Authors:** Ikki Kobayashi, Kenji Mochizuki, Jun Teramoto, Sousuke Imamura, Kazuhiro Takaya, Akira Ishihama, Tomohiro Shimada

**Affiliations:** 1School of Agriculture, Meiji University, Kawasaki 214-8571, Kanagawa, Japan; cyano7942@gmail.com; 2Micro-Nano Technology Research Center, Hosei University, Koganei 184-0003, Tokyo, Japan; k.mochi.0816@gmail.com (K.M.); jyunne@hotmail.com (J.T.); 3Space Environment and Energy Laboratories, Nippon Telegraph and Telephone Corporation, Musashino-shi 180-8585, Tokyo, Japan; sousuke.imamura.hp@hco.ntt.co.jp (S.I.); kazuhiro.takaya.bm@hco.ntt.co.jp (K.T.)

**Keywords:** transcription factor, stationary phase, biofilm formation, hydrogen peroxide sensitivity, genomic SELEX (gSELEX), *Escherichia* *coli*

## Abstract

Understanding the functional information of all genes and the biological mechanism based on the comprehensive genome regulation mechanism is an important task in life science. YgfI is an uncharacterized LysR family transcription factor in *Escherichia coli*. To identify the function of YgfI, the genomic SELEX (gSELEX) screening was performed for YgfI regulation targets on the *E. coli* genome. In addition, regulatory and phenotypic analyses were performed. A total of 10 loci on the *E. coli* genome were identified as the regulatory targets of YgfI with the YgfI binding activity. These predicted YgfI target genes were involved in biofilm formation, hydrogen peroxide resistance, and antibiotic resistance, many of which were expressed in the stationary phase. The TCAGATTTTGC sequence was identified as an YgfI box in in vitro gel shift assay and DNase-I footprinting assays. RT-qPCR analysis in vivo revealed that the expression of YgfI increased in the stationary phase. Physiological analyses suggested the participation of YgfI in biofilm formation and an increase in the tolerability against hydrogen peroxide. In summary, we propose to rename *ygfI* as *srsR* (a stress-response regulator in stationary phase).

## 1. Introduction

The growth of bacteria in a natural environment is often limited; so, within these environments, they subsequently enter a stationary phase. The long-term survival of bacteria under stressful conditions depends on the establishment of various stress defense systems. For example, stationary phase adaptation of bacteria is accompanied by marked changes in their gene expression patterns [1,2,3]. *Escherichia coli*, as an example, contains a total of approximately 4500 protein-coding sequences on its genome [4,5]. The genes required for the response to various stresses and for survival in the stationary phase have been identified by comprehensive analyses using gene deletion or overexpression strains [6,7]. At present, however, even within the model bacterium *E. coli*, the function of one-fourth of the species is still unknown because most stress-response genes are not expressed within laboratory conditions. Accordingly, the full set of regulators involved in the expression of stress conditions during the stationary phase has not yet been identified. In the absence of knowledge of transcription factors (TFs) and conditions affecting the expression of regulatory functions, the ordinary in vivo approach is not useful in identifying regulatory targets of hitherto uncharacterized TFs, as TFs are not always expressed [8]. At present, the function of one-fifth of *E. coli* TFs remains unidentified [9].

To rapidly identify the regulation targets by uncharacterized TFs, we developed an improved system of genomic systematic evolution of ligands using exponential enrichment (SELEX) [10,11], and successfully identified regulation targets for several TFs [9,12]. This gSELEX screening system using the reaction in vitro between purified test TF proteins and genome DNA fragments is especially useful in identifying regulatory targets of hitherto uncharacterized TFs, of which their expression conditions are unknown. Using the gSELEX system, we identified the regulation targets and regulatory roles for some unknown Y-TFs, including YagI (renamed as XynR), YbaO (renamed as DecR), YbiH (renamed as CecR), YbjK (renamed as RcdA), YcdC (renamed as RutR), YcjZ (renamed as PgrR), YdeO (renamed as PhhR), YdcN (renamed as SutR), YdfH (renamed as RspR), YdhM (renamed as NemR), YeaM (renamed as NimR), YedW (renamed as HprR), YgiP (renamed as Dan), YiaJ (renamed as PlaR), YiaU (renamed as CsuR), and YihW (renamed as CsqR) [9,13,14,15,16,17]. The gSELEX analysis can also lessen the difficulty of in vivo analysis because a set of regulatory proteins involved in the regulation of a single promoter competes to bind to the overlapping DNA sites [18]. Although difficulties exist, bacterial DNA-binding TFs bind to DNA and regulate nearby genes; thus, predicting regulation target promoters, genes, and operons is possible based on the location of recognition sequences by test TFs.

As part of our research strategy to understand the function of the comprehensive set of *E. coli* TFs, we sought to characterize YgfI in this study, which has been recognized as one of the uncharacterized TFs in *E. coli* and belongs to the LysR family [9]. A systematic search was performed to identify target genes controlled by YgfI using the gSELEX-chip system [11]. As a result, a total of 10 strong YgfI-binding sites were identified throughout the *E. coli* genome. The predicted YgfI target genes are involved in stress responses such as biofilm formation, antibiotic resistance, and hydrogen peroxide resistance. Genes involved in these stress responses also facilitate adaptations to the stationary phase. The transcriptional regulation and physiological roles of YgfI were experimentally examined. As a result, YgfI activated a set of genes in the stationary phase. We therefore propose renaming YgfI as SrsR (a *s*tress-*r*esponse regulator in the *s*tationary phase).

## 2. Results

### 2.1. Regulation Targets of SrsR per gSELEX-Chip Screening

To identify the set of binding sequences for SrsR, we performed gSELEX screening using an improved method [10,11]. In brief, purified His-tagged SrsR was mixed with a collection of *E. coli* genome fragments of 200–300 bp in length, and SrsR-bound DNA fragments were affinity-isolated. The original substrate mixture of original genomic DNA fragments formed smear bands on PAGE, but after six cycles of gSELEX, DNA fragments with a high affinity for SrsR were enriched, forming sharper bands on PAGE gels.

To identify the comprehensive set of targets directly controlled by SrsR, gSELEX fragments were labeled with Cy3, whereas the original genomic DNA library was labeled with Cy5. A mixture of fluorescently labeled samples was hybridized to an *E. coli* DNA tiling microarray [19,20]. The ratio of the fluorescence intensity bound to each probe between the SrsR sample and the original library DNA was measured and plotted against the corresponding position along the *E. coli* K-12 genome. The extent of SrsR binding correlated with its affinity for the SrsR protein. After setting the cutoff level at 200, a total of seven peaks were identified in the gSELEX-chip pattern, and three additional peaks were identified by decreasing the cutoff level to 150 (Figure 1). The highest peak was identified in the spacer region of the *yfdO* and *yfdP* genes, which are located in the CPS-53 prophage. Based on the criteria that prokaryotic TF-binding sites are located upstream of the regulatory target genes [12,21], a total of 10 genes (*ydfO, yfdP, fucA, fucP, dinJ, yafL, bluR, paaZ, paaA*, and *cdgI*) were predicted to be potential regulatory targets of SrsR (Table 1). Because some of these genes form operons, the total number of regulatory target genes was estimated to be 30 (Table 1).

Genes involved in biofilm formation, such as *bluR* (regulator of the *ycgZ-ymgA-ariR-ymgC* operon, a putative two-component system connector involved in biofilm formation), *cdgI* (putative c-di-GMP binding protein), *fucAO,* and *fucK* (fucose metabolism), are included as an abundant group in the list of predicted targets of SrsR (see below for biofilm formation assay). In addition, *yafQ* (toxin of the YafQ-DinJ toxin-antitoxin system) [22] and *yfdO* (prophage gene) [7] were reportedly involved in antibiotic resistance. Notably, some of these genes are expressed during the stationary phase [23,24,25]. These perceptions of SrsR target genes suggest that SrsR is involved in the stress response during the stationary phase.

### 2.2. Binding In Vitro of SrsR to the Target Sequences

The binding in vitro of purified SrsR to sequences isolated by the gSELEX screening was examined by PAGE analysis to confirm the formation of SrsR–DNA complexes. Upon increasing the SrsR addition, DNA fragments from all the major peaks of the gSELEX-chip formed SrsR–DNA complexes in an SrsR concentration-dependent manner (Figure 2a–j). Several SrsR–DNA complexes were observed for the *yfdO/yfdP* spacer probe (Figure 2h), indicating that SrsR binds to several sites in the *yfdO/yfdP* intergenic region. Generally, the target promoter, which carries multiple binding sites for the test regulator, shows highly stable TF–probe complexes. Thus, it is reasonable that the *yfdO/yfdP* spacer region showed the highest binding affinity for gSELEX screening (Figure 1). In contrast, the *lacUV5* promoter region, a reference DNA added as a negative control, did not form SrsR–DNA complexes under the same condition (Figure 2k). These results indicate the specific binding of SrsR to all 10 SrsR target sequences (Table 1).

To identify the binding sequence of SrsR, we performed a DNase-I footprinting assay using the *yfdP/yfdO* spacer as a probe. A 66 bp segment was protected from DNase-I digestion in the presence of SrsR (Figure 3a). Within this segment, four sets of 11 bp sequence TCAGATTTTGC were identified (Figure 3a). Next, a collection of 500 bp sequences from the set of 10 SrsR targets was analyzed using the MEME program [26] to identify the SrsR box sequence. We identified the same 11 bp sequence found in the *yfdP/yfdO* spacer region by the DNase-I footprinting assay (Table 1 and Figure 3b). Multiple and highly conserved SrsR boxes were found only in the *yfdP/yfdO* spacer (Table 1), in accordance with the presence of several SrsR–probe complexes observed in the gel shift assay (Figure 2h). Thus, we concluded that this 11 bp SrsR box sequence is required to ensure tight binding of SrsR.

### 2.3. Growth-Dependent Expression Level of srsR Gene

Some SrsR targets have reportedly been induced in the stationary phase, including *bluR* and its targets *ycgZ-ymgA-ariR-ymgC* [23,27], *cdgI* [24], and *dinJ* [25]. RT-qPCR analysis was, therefore, performed to measure SrsR expression levels over time. For this purpose, total RNAs were purified over time from cultured cells grown in an LB medium, and the expression level of *srsR* was measured. As a result, the mRNA level of *srsR* increased during the transition from the exponential growth phase to the stationary phase, finally reaching approximately 25 times higher levels than those of the exponential phase (Figure 4). This result indicates that SrsR is influential during the growth transition to the stationary phase and regulates a set of target genes in the stationary phase.

### 2.4. Regulatory Role of Srsr in Expression of the Target Genes In Vivo

To examine the possible influence of SrsR on the target promoters detected in vitro based on SrsR-binding sites, we performed RT-qPCR analysis to determine mRNA levels in vivo for each of the predicted SrsR target genes in the presence, absence, or overexpression of SrsR (Figure 5). Total RNA was prepared from cells of wild-type *E. coli* K-12, the wild-type strain harboring the SrsR overexpression vector or the control empty vector, and an *srsR*-deleted mutant; all were grown in an LB medium, and the mRNA levels of individual *srsR* target genes were measured. Total RNA was prepared at both exponential and stationary phases and subjected to RT-qPCR. Except for the *yfdONMLK* and *yfdPQ* operons, the mRNA levels of targets could not be detected in all tested strains from the exponential phase cells (data not shown). In contrast, RNA of all the tested genes were detected in stationary phase cells. However, RNA decreased in the *srsR* mutant strain compared with the wild-type strain (Figure 5, black bar), whereas RNA from all the tested genes was increased through the SrsR overexpression (Figure 5, white bar). These results indicate that the tested genes influenced the stress response; thus, the members of the SrsR regulon participated in the stress response in the stationary phase. In summary, we propose that SrsR activates a set of genes in the stationary phase.

### 2.5. Regulatory Role of SrsR in Biofilm Formation

The qRT-PCR analysis suggested that SrsR influences the activation of a set of genes involved in biofilm formation, including *bluR* and its targets *ymgA-ariR-ymgC* [23,27,28], *cdgI* [29], *fucAO* [30], and *fucK* [31]. The involvement of SrsR as a key regulator of biofilm formation was then confirmed using the crystal violet staining assay [32]. Under the test conditions, the level of biofilm formation increased in the mutant JW5476 lacking *srsR* (Figure 6). In contrast, the wild-type strain harboring the SrsR overexpression vector showed markedly decreased biofilm formation, whereas the wild-type strain harboring the empty vector had no effect on biofilm formation (Figure 6).

The level of biofilm formation has reportedly increased in gene deletion strains of *ymgA, ariR*, and *ymgC,* which are repressed by BluR [23,27], and also *cdgI* [29] grown in LB medium. Moreover, strains overexpressing YmgA or YmgB activate the RcsCDB repressor, which influences the formation of curli and fimbriae [23]. The mRNA levels of *bluR* and *cdgI* decreased 1.6 and 5.9-fold in the *srsR* mutant strain and increased 7.2 and 4.1-fold following the overexpression of SrsR, respectively (Figure 5). Together, these findings support the conclusion that SrsR influences the repression of biofilm formation.

### 2.6. Regulatory Role of the SrsR in Hydrogen Peroxide Sensitivity

The highest level of SrsR-binding activity was observed in the *yfdO/yfdP* intergenic region (Figure 1) within the CPS-53 prophage locus. Deletion of the prophage CPS-53 strain has reportedly increased hydrogen peroxide (H_2_O_2_) sensitivity [6]. Moreover, this report described that the deletion of *yfdK, yfdO,* and *yfdS* in the CPS-53 prophage reduced viability against oxidative stress induced by H_2_O_2_. In accordance with our findings, the mRNA level of the *yfdONMLK* operon in the stationary phase decreased by 3.7-fold as detected by *yfdO* mRNA and 15.0-fold as per *yfdK* mRNA in the *srsR* mutant strain; however, it increased by 10.7-fold as detected by *yfdO* mRNA and 1052-fold as per *yfdK* mRNA in the SrsR overexpression strain (Figure 5).

Based on these findings, we tested the role of SrsR in H_2_O_2_ resistance. Both the *E. coli* wild-type strain and its single-gene *srsR*-deletion strain were grown in LB medium until the exponential phase, and then exposed to 30 mM H_2_O_2_ (Figure 7a). Following a 15 min exposure to H_2_O_2_, the level of cell survival in the wild-type strain was 1.3%, whereas that of the *srsR* mutant strain decreased to 0.18%, thereby indicating that the presence of SrsR is needed for H_2_O_2_ resistance. The effect of SrsR was confirmed by testing the SrsR overexpression vector. The wild-type strain harboring the vector led increased cell viability to 19% under the same conditions, but no effect occurred with the empty vector pCA24N*Δgfp* (Figure 7a).

To confirm the regulatory role of SrsR in H_2_O_2_ resistance, mRNA levels of the wild-type strain and *srsR* mutant strain in the exponential phase were determined using Northern blot analysis. The *yfdK* probe should be included in the *yfdONMLK* operon transcript, which is shorter than the 2904-nucleotide 23S rRNA on the membrane (Figure 7b). The level of the *yfdONMLK* transcript was detected in the wild-type strain harboring the SrsR overexpression vector and was further increased by the addition of IPTG for SrsR induction. Since Northern blotting analysis was not sensitive enough to detect the transcript levels of the operon in wild-type and *srsR* deletion strains, next, we performed RT-qPCR analysis. RT-qPCR analysis showed the mRNA level of the *yfdONMLK* operon in the exponential phase decreased by 9.5-fold as detected by *yfdO* mRNA and 2.8-fold as per *yfdK* mRNA in the *srsR* mutant strain, whereas it increased by 12.4-fold as detected by *yfdO* mRNA and 2.2-fold as per *yfdK* mRNA in the SrsR overexpression strain (Figure 7c). These results suggest that SrsR activates the *yfdONMLK* operon not only during the stationary phase but also in the exponential phase. The H_2_O_2_ sensitivity of the *yfdK* mutant decreased by approximately 103 times (Figure 7a). Similarly, the single-gene *yfdO* and *yfdS* deletion strains exhibited the same level of sensitivity (data not shown), thus confirming the influence of the *yfdONMLK* operon in hydrogen peroxide sensitivity. This finding is in accordance with previously published observations [6], while the *srsR* mutant showed a 10-fold increase in sensitivity compared with the wild-type strain (Figure 7a). Although less severe than the *yfdK* deletion mutant, increased sensitivity to oxidative stress was also observed in the *srsR* mutant, a regulator of the *yfdKONMLK* operon, suggesting that SrsR activates the operon and increases the tolerability against hydrogen peroxide.

## 3. Discussion

A set of stress-response genes exists in bacterial genomes [2,3]. Some of these genes are expressed for survival during the stationary phase. To detail the mechanisms of stress adaptation, it is essential to clarify the regulatory roles of stress-response genes. In this study, we identified that the hitherto uncharacterized *E. coli* transcription factor SrsR (renamed from YgfI) is a local regulator of a group of stress-response genes, including biofilm formation, hydrogen peroxide sensitivity, drug resistance, and cold shock stress. The expression level of SrsR increased in the stationary phase and during activation of the set of stress-response genes; we, therefore, propose to rename *ygfI* to *srsR* (stress-response regulator in stationary phase).

Several genes have been identified in biofilm formation in *E. coli* K-12 [33,34,35], whereas only a limited number of regulators have been identified in the deceleration of biofilm formation. One thoroughly characterized regulator inhibiting biofilm formation influences the repression of the *csgDEFG* operon, which encodes a set of genes involved in the formation of curli fimbriae. The level of biofilm formation decreases, owing to the repression of the *csgD* promoter, where CpxR, FliZ, H-NS, MqsA, BtsR, RcsAB, and RstA act as repressors [18,36]. Another deceleration system of biofilm formation includes BluR as a repressor of the *ycgZ-ymgA-ariR-ymgC* operon, which inhibits biofilm formation via the cascade for activating *rcsDCB* and repressing curli and fimbriae formation [23]. Similar to these transcription factors, SrsR regulates a set of genes including *bluR* as a new inhibitor of biofilm formation during the stationary phase (Figure 6).

While SrsR suppressed biofilm formation, SrsR increased the tolerability against hydrogen peroxide (Figure 7). *E. coli* carries several regulators for activating a set of genes for oxidative stress, including OxyR, SoxR, and SoxS [37]. Recently, it was reported that Cys25 of OxyR, which is *S*-nitrosylation in planktonic cells, plays an important role in suppressing biofilm formation [38]. Such transcriptional regulation by SrsR and OxyR suggests the importance of switching between the biofilm and planktonic lifestyles as well as conferring oxidative stress tolerance to planktonic cells in bacterial adaptation to the stressful environment.

For Enterobacteriaceae, including *E. coli*, survival and infection in the host gut is important to their survival strategy. In contrast, the host produces reactive oxygen species such as hydrogen peroxide to eliminate pathogens [39,40]. This study showed that not only the *yfdONMLK* operon, which is significant for oxidative stress tolerance, but also the *fucAO* operon and *fucPIKUR* operon, which are required for survival in the host gut [30,31], were activated by SrsR (Figure 5). These findings suggest that SrsR may be a crucial transcription factor for survival strategies of *E. coli* in the host gut.

The SrsR regulon may also be involved in adaptation to nitrogen starvation conditions, since it had been reported that *srsR* is repressed by Nac (nitrogen assimilation control), a transcription factor that is expressed under nitrogen starvation conditions [41]. SrsR belongs to the LysR family, which includes the maximum number (47 members) of TFs in *E. coli* [9]. The TF of the LysR family contains a DNA-binding domain at the C-terminus and a co-inducer-binding domain at the N-terminus. Further studies are needed to identify a co-inducer to elucidate further physiological functions of SrsR.

## 4. Materials and Methods

### 4.1. Bacterial Strains and Plasmids

*Escherichia coli* K-12 W3110 type-A [42] was used as the DNA source to construct the SrsR expression plasmid. The *E. coli* K-12 W3110 type-A genome was also used to construct the DNA library needed for gSELEX screening. *E. coli* DH5α cells were used to perform plasmid amplification, and *E. coli* BL21 (DE3) cells were used as the SrsR expression. *E. coli* BW25113 [43], its single-gene knockout mutants (JW5476 for *srsR* and JW2350 for *yfdK*) [44], and the expression plasmid from the ASKA clone library were obtained from the *E. coli* Stock Center (National Bio-Resource Center, Saitama, Japan). Plasmid pPET21 was used to construct the SrsR expression plasmid pSrsR. Cells were grown in an LB medium at 37 °C with constant shaking at 150 rpm. When necessary, 20 μg mL^−1^ kanamycin or 30 μg mL^−1^ chloramphenicol was added to the medium. When observing the effect of SrsR overexpression, IPTG at a final concentration of 0.5 mM was added from the start of inoculation. Cell growth was monitored by measuring turbidity at 600 nm.

### 4.2. Purification of SrsR Protein

Plasmid pSrsR for expressing and purifying SrsR was constructed according to the standard procedure [10]. In brief, SrsR coding sequences were PCR-purified using the *E. coli* K-12 W3110 genome DNA as a template and inserted into the pET21a (+) vector (Novagen, Darmstadt, German) between NdeI and NotI sites, leading to the construction of pSrsR. The expression plasmid, pSrsR, was transformed into *E. coli* BL21 (DE3) cells. Transformants were grown in an LB medium, and SrsR was expressed by adding IPTG to the middle of the exponential phase. The SrsR protein was purified by affinity purification using a Ni-nitrilotriacetic acid (NTA) agarose column. The affinity-purified SrsR protein was stored, frozen, in the storage buffer at −80 °C until use. Protein purity was greater than 95%, as determined by SDS-PAGE.

### 4.3. Genomic SELEX (gSELEX) Screening of SrsR-Binding Sequences

The gSELEX screening was performed as previously described [10,11]. Briefly, a mixture of DNA fragments of the *E. coli* K-12 W3110 genome was prepared by sonication of purified genomic DNA and cloned into a multi-copy plasmid, pBR322. For each gSELEX screening, the DNA mixture was regenerated by PCR. For gSELEX screening, 5 pmol of the mixture of DNA fragments and 10 pmol SrsR were mixed in a binding buffer (10 mM Tris-HCl, pH 7.8 at 4 °C, 3 mM magnesium acetate, 150 mM NaCl, and 1.25 mg/mL bovine serum albumin). The SELEX cycle was repeated six times to enrich the SrsR-binding sequences. Mapping of SELEX fragments along the *E. coli* genome was also performed using a gSELEX-chip system with a 43,450-feature DNA microarray [45]. The genomic SELEX sample obtained using SrsR was labeled with Cy3, whereas the original genomic DNA library was labeled with Cy5. Following hybridization of samples to the DNA tilling array (Agilent technology, Santa Clara, CA, USA), the Cy3/Cy5 ratio was measured, and the peaks of the scanned patterns were plotted against the positions of the DNA probes along the *E. coli* K-12 genome.

### 4.4. Gel Shift Assay

A gel shift assay was performed according to the standard procedure [46]. Probes for the SrsR-binding target sequences were generated by PCR amplification using a pair of primers (Appendix A) and Ex Taq DNA polymerase (TaKaRa, Kyoto, Japan). A mixture of each probe and SrsR was then incubated at 37 °C for 30 min in the gel shift buffer. After adding the DNA loading solution, the mixture was directly subjected to PAGE. DNA in the gels was stained with GelRed (Biotium, Fremont, California, USA) and detected using LuminoGraph I (Atto, Tokyo, Japan).

### 4.5. DNase-I Footprinting Assay

A DNase-I footprinting assay was performed under standard reaction conditions [47]. Probes for the *yfdP/yfdO* sequence were generated by PCR amplification using a pair of primers (Appendix A) and Ex Taq DNA polymerase (TaKaRa). For footprinting, 1.0 pmol of FITC-labeled probes was incubated at 37 °C for 30 min with various concentrations of purified SrsR in 25 μL of a solution containing 10 mM Tris-HCl (pH 7.8), 150 mM NaCl, 3 mM magnesium acetate, 5 mM CaCl_2_, and 25 mg/mL bovine serum albumin. Following incubation, DNA digestion was initiated by adding 5 ng DNase-I (TaKaRa). Then, following digestion for 30 s at 25 °C, the reaction was terminated through an addition of 25 μL of phenol. DNA was precipitated from the aqueous layer by ethanol, dissolved in a formamide dye solution, and analyzed by electrophoresis on a 6% polyacrylamide gel containing 8 M urea.

### 4.6. Consensus Sequence Analysis

To analyze the SrsR-binding sequence, an SrsR-binding sequence set identified by the gSELEX-chip was analyzed using the MEME program [26]. Sequences were aligned, and a consensus sequence logo was created by WEBLOGO (Available online: http://weblogo.berkeley.edu/logo.cgi (accessed on 1 June 2021)).

### 4.7. RT-qPCR Analysis

RT-qPCR was performed according to standard procedure [48]. *E. coli* cells were inoculated into an LB medium at 37 °C under aeration, with constant shaking at 150 rpm. After inoculation for 24 h, the total RNA was extracted using an ISOGEN solution (Nippon Gene). Total RNAs was transcribed to cDNA with random primers using the THUNDERBIRD SYBR qPCR RT Set (TOYOBO, Osaka, Japan). Quantitative PCR (qPCR) was conducted using the THUNDERBIRD SYBR qPCR Mix (TOYOBO) and a LightCycler 96 system (Roche, Barsel, Switzerland). The primer pairs used are described in Appendix A. The cDNA templates were serially diluted 4-fold and used in the qPCR assays. The qPCR mixtures, each containing 10 μL of THUNDERBIRD SYBR qPCR Mix (TOYOBO), 1 μL of each primer (5 μM stock), 7 μL of water, and 1 μL of cDNA, were amplified under the following thermal cycle conditions: 95 °C treatment for 2 min, at 45 cycles of 10 s at 95 °C, and 20 s at 55 °C, and then incubation for 20 s at 72 °C. The 16S rRNA expression level was used to normalize the varying levels of the test samples, and the relative expression levels were quantified using the Relative Quantification software provided by Roche. The results are presented as the average of three independent experiments.

### 4.8. Biofilm Assay

The crystal violet staining method was used as previously described in another study [32]. *E. coli* cells were grown in an LB medium at 37 °C in a 2.0 mL tube. Following incubation for 24 h, planktonic cells were discarded, the tube was washed twice with PBS (-) and subsequently stained with 0.1% crystal violet for 20 min at room temperature. After extensive washing with H_2_O, biofilm-bound crystal violet was extracted with 500 μL of 70% ethanol and measured at OD_595_.

### 4.9. Hydrogen Peroxide Test

*E. coli* cells were inoculated in an LB medium at 37 °C under aeration with constant shaking at 150 rpm until the OD_600_ reached 0.4, and then hydrogen peroxide was added to the culture to a final concentration of 30 mM. Following this addition, incubation continued for 15 min, and then the cells were diluted and spread on LB agar plates. Untreated hydrogen peroxide-treated cells were also plated. The plates were incubated at 37 °C overnight, and then the number of colonies on the plates was counted. The survival rate was calculated as the ratio of colony numbers between hydrogen peroxide-treated cells and untreated cells.

### 4.10. Northern Blot Analysis

Total RNAs were extracted from the exponential phase *E. coli* cells (OD_600_ = 0.4) by ISOGEN solution (Nippon Gene). RNA purity was checked by electrophoresis on 1.5% agarose gel with formaldehyde present, followed by staining with GelRed. Northern blot analysis was performed as described in previous research [49]. DIG-labeled probes were prepared by PCR amplification using W3110 genomic DNA (50 ng) as a template, with a pair of primers (Appendix A), DIG-11-dUTP (Roche) and dNTP, as substrates, gene-specific forward and reverse primers, and Ex Taq DNA polymerase. Total RNAs (3 μg) were incubated in a formaldehyde-MOPS (morpholinepropanesulfonic acid) gel-loading buffer for 10 min at 65 °C for denaturation, subjected to electrophoresis on formaldehyde-containing 1.5% agarose gel, and then transferred onto nylon membrane (Roche). Hybridization was performed using a DIG easy Hyb system (Roche) at 50 °C overnight with a DIG-labeled probe. To detect the DIG-labeled probe, the membrane was treated with anti-DIG-AP Fab fragments and CDP-Star (Roche), and the image was scanned using LuminoGraph I (Atto).

## Figures and Tables

**Figure 1 ijms-23-06055-f001:**
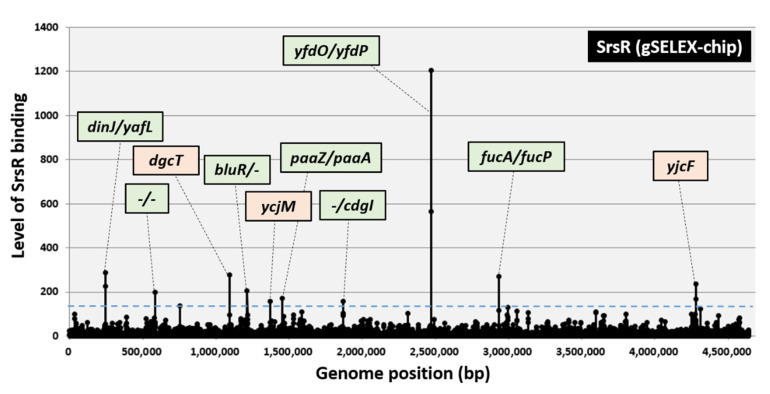
Identification of SrsR-binding sites on the *E. coli* K-12 genome by gSELEX-chip. The gSELEX screening of DNA-binding sequences was performed for SrsR (renamed YgfI), a yet uncharacterized LysR family TF of *E. coli*, using purified C-terminal His-tagged SrsR and a library of DNA segments from the *E. coli* K-12 W3110 genome. Following gSELEX, a collection of DNA fragments was subjected to gSELEX-chip analysis using the tiling array of the *E. coli* K-12 genome. The cutoff level of 150 is shown by a blue line, and the list of all SrsR-binding sites from setting this cutoff level is given in Table 1. Peaks shown in green represent the SrsR-binding sites inside spacer regions, whereas peaks shown in orange represent the SrsR-binding sites inside ORFs.

**Figure 2 ijms-23-06055-f002:**
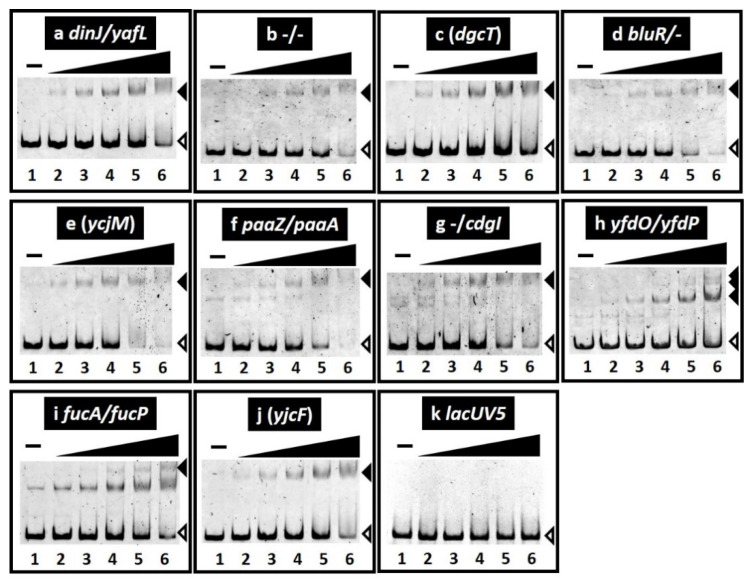
Gel shift assay of SrsR–DNA complex formation. (**a**–**k**) Purified SrsR was mixed with 0.1 pmol of each target DNA probe to the SrsR-binding regions shown in Figure 1. SrsR added were (in pmol): lane 1, 0; lane 2, 0.125; lane 3, 0.25; lane 4, 0.5; lane 5, 1; and lane 6, 2. Filled triangles indicate the SrsR–DNA probe complex whereas open triangles indicate free probes.

**Figure 3 ijms-23-06055-f003:**
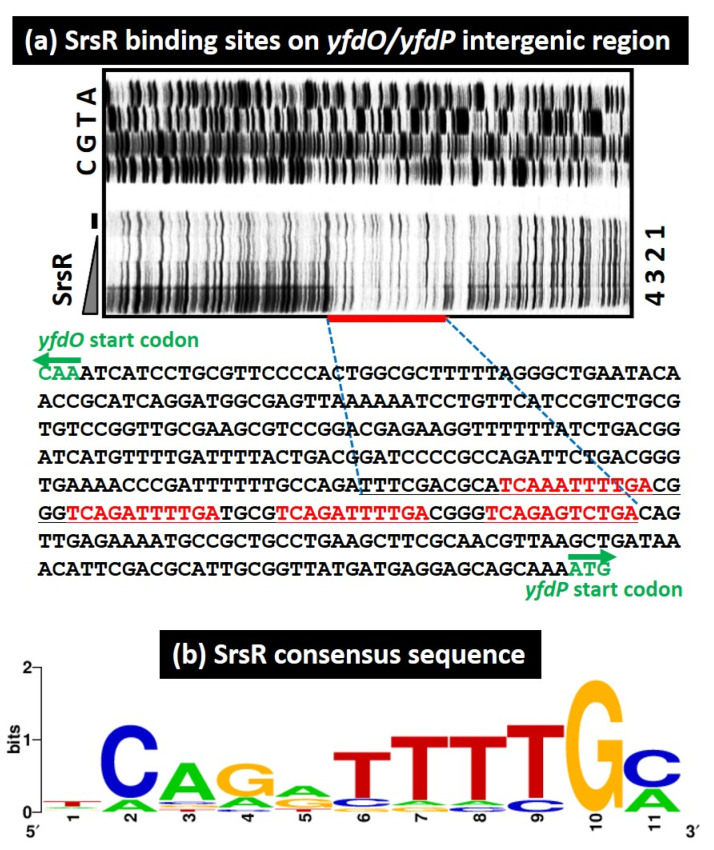
SrsR binding sequence. (**a**) DNase-I footprinting assay for identifying the SrsR binding site on the *yfdP/yfdO* intergenic region. Mixtures of FITC-labeled DNA probe (1.0 pmol) of the *yfdP/yfdO* spacer sequence and increasing amounts of purified SrsR (0, 10, 20, 40 μM) were subjected to the DNase-I treatment. The sequence shown includes those between the initiation codon of the *yfdO* gene and the initiation codon of the *yfdP* gene. The red bar indicates the DNase-I protected region (SrsR box) by SrsR. The DNase-I protected region by SrsR is shown in black. (**b**) Consensus sequence of the SrsR box. Sequences of all the probes with high-level binding of SrsR were analyzed using THE MEME Suite (Available online: https://meme-suite.org/meme/ (accessed on 1 June 2021)) (see Table 1). WEBLOGO (Available online: http://weblogo.berkeley.edu/logo.cgi (accessed on 1 June 2021)) was used to perform matrix construction.

**Figure 4 ijms-23-06055-f004:**
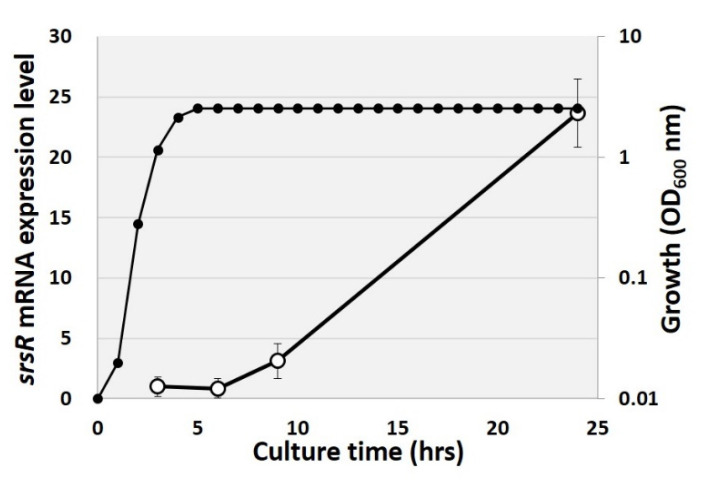
Changing of *srsR* mRNA level in growth curve. Total RNA was purified from the *E. coli* wild-type strain at 3, 6, 9, and 24 h of cultivation, and the cell growth was measured by OD_600_ nm every 1 h (closed symbol). The *srsR* mRNA level was compared by setting the ratio against 16S rRNA as an internal control (open symbol).

**Figure 5 ijms-23-06055-f005:**
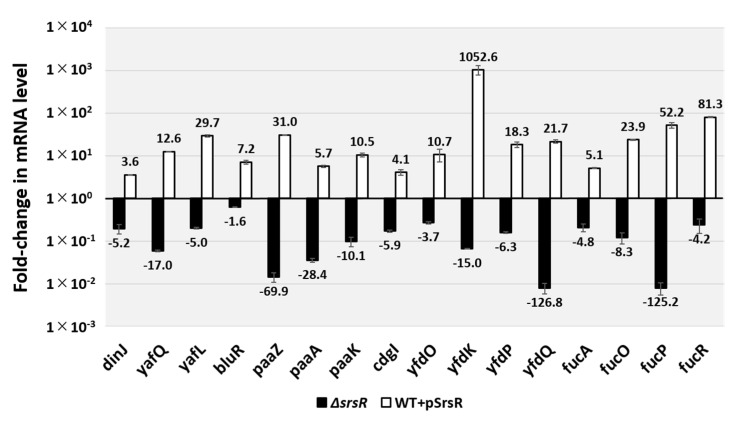
In vivo influence of *srsR* deletion or SrsR overexpression on transcription level of the SrsR targets by RT-qPCR. *E. coli* wild-type BW25113, its *srsR*-deleted mutant JW5476, the wild-type strain harboring the SrsR overexpression vector, and its empty vector were grown in the LB medium at 37 °C. Total RNA was prepared from each strain at 24 h cultured cells and subjected to RT-qPCR analysis. RT-qPCR was repeated three times, and the average values are shown. The *y*-axis represents the relative level of mRNA of each SrsR target gene between the wild-type and the *srsR* mutant (black bar) as well the relative level between the wild-type harboring SrsR overexpression vector and empty vector (white bar); the ratio of 16S rRNA was set as an internal control between the compared strains. Each experiment was repeated at least three times, and the average means are shown.

**Figure 6 ijms-23-06055-f006:**
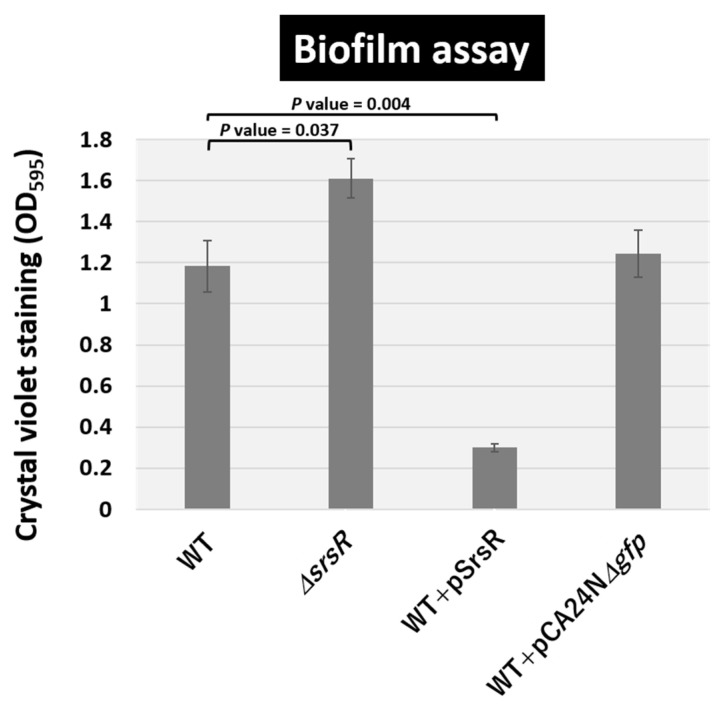
Level of biofilm formation. *E. coli* K-12 wild-type, the isogenic *srsR* mutant JW5476, the wild-type strain harboring SrsR overexpression vector, and its empty vector were grown in LB medium at 37 °C for 24 h in a plastic tube. The level of biofilm formation was measured by crystal violet staining. The relative level of biofilm represents the mean ± SD of three experiments. Statistically significant differences were obtained using the Student’s t-test with multiple comparisons’ correction and are represented by line over bars with associated *p* values.

**Figure 7 ijms-23-06055-f007:**
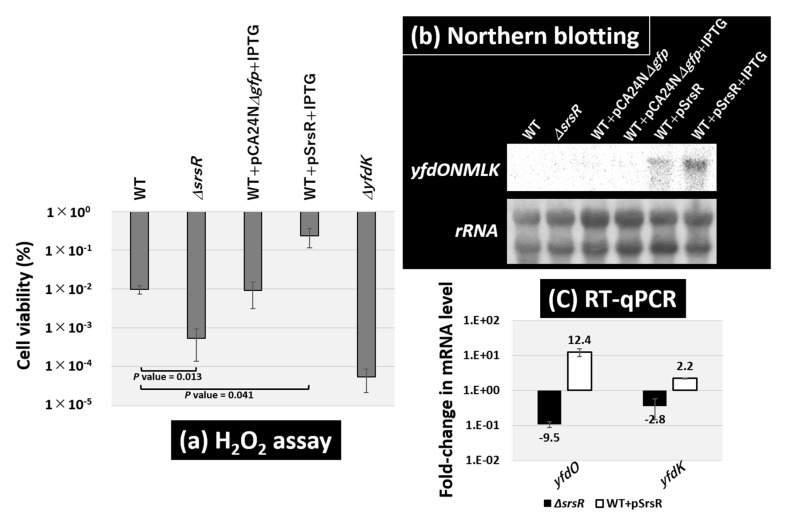
Sensitivity test against hydrogen peroxide. (**a**) Survival test for the *E. coli* wild-type BW25113, its *srsR*-deleted mutant JW5476, the wild-type strain harboring SrsR overexpression vector, its empty vector, and *yfdK* mutant JW2350. Each strain was inoculated until the OD_600_ reached 0.4 (exponential phase) and then subjected to 30 mM H_2_O_2_ for 15 min. The relative level of sensitivity represents the mean ± SD of three experiments. Statistically significant differences were obtained using the Student’s t-test with multiple comparisons’ correction and are represented by line under bars with associated *p* values. (**b**) *E. coli* wild-type BW25113 and its *srsR*-deleted mutant JW5476, the wild-type strain harboring its empty vector, and SrsR overexpression vector with or without IPTG were grown in an LB medium at 37 °C until the OD_600_ reached 0.4 (exponential phase). Total RNA was prepared from both wild-type and the *srsR* mutant and subjected to Northern blot analysis. The mRNA of the *yfdONMLK* transcription unit detected with *yfdK* DIG-labeled hybridization probes is shown. The amounts of total RNA analyzed were examined by measuring the intensity of ribosomal RNAs. (**c**) The same RNAs as in panel (**b**) were subjected to RT-qPCR analysis. The *y*-axis represents the relative level of mRNA of *yfdO* and *yfdK* between the wild-type and the *srsR* mutant (black bar) as well the relative level between the wild-type harboring SrsR overexpression vector and empty vector (white bar); the ratio of 16S rRNA was set as an internal control between the compared strains. Each experiment was repeated at least three times, and the average means are shown.

**Table 1 ijms-23-06055-t001:** SrsR-binding sites on the *E.coli* genome.

	Peak Position (bp)	Intensity	Function	Operon	Gene	D	SrsR Site	D	Gene	Operon	Function	SrsR Box TCAGATTTTGC	Conservation
**1**	**246,530**	**288**	**Antitoxin/DNA-binding transcriptional repressor**	** *dinJ-yafQ* **	** *dinJ* **	<		>	** *yafL* **	** *yafL* **	**predicted lipoprotein and C40 family peptidase**	**aaAGgcaTTGC**	**6/11**
**2**	**583,766**	**200**			*appY*	>		<	*ompT*			**TCcatTTTTGC**	**8/11**
**3**	**1,092,430**	**278**			*pgaA*	<	*dgcT*	<	*insEF*			**cCAGgcgTTGa**	**6/11**
**4**	**1,213,450**	**207**	**Transcriptional repressor**	** *bluR* **	** *bluR* **	<		<	*ycgF*			**gCAGgTTTTGC**	**9/11**
**5**	**1,368,672**	**158**			*pspE*	>	*ycjM*	>	*ycjN*			**TCActTTTTGC**	**9/11**
**6**	**1,451,756**	**170**	**Oxepin-CoA hydrolase/3-oxo-5,6-dehydrosuberyl-CoA semialdehyde dehydrogenase**	** *paaZ* **	** *paaZ* **	<		>	** *paaA* **	** *paaABCDEFGHIJK* **	**predicted ring 1,2-phenylacetyl-CoA epoxidase subunit**	**aCgGATTTcGC**	**8/11**
**7**	**1,868,366**	**158**			*yeaH*	>		>	** *cdgI* **	** *cdgI* **	**c-di-GMP binding protein**	**aaAGgTTaTGC**	**7/11**
**8**	**2,471,464**	**1206**	**CPS-53 (KpLE1) prophage; predicted defective phage replication protein O**	** *yfdONMLK* **	** *yfdO* **	<		>	** *yfdP* **	** *yfdPQ* **	**CPS-53 (KpLE1) prophage; predicted protein**	**TCAaATTTTGa, TCAGATTTTGa, TCAGATTTTGa, TCAGAgTcTGa**	**9/11, 10/11, 10/11, 8/11**
**9**	**2,931,960**	**270**	**L-fuculose-phosphate aldolase**	** *fucAO* **	** *fucA* **	<		>	** *fucP* **	** *fucPIKUR* **	**FucP fucose MFS transporter**	**gCtGgTTTTGC**	**8/11**
**10**	**4,280,360**	**235**			*yjcE*	>	*yjcF*	<	*actP*			**TCAaATTTcGC**	**9/11**

gSELEX was used to search for the binding sites in SrsR. Ten binding sites were identified by setting the cutoff level to 150. Column ‘SrsR site’ shown in green represent the SrsR-binding sites inside spacer regions, whereas shown in orange represent the SrsR-binding sites inside ORFs. Column ‘D’ shows the direction of the transcription unit. The potential target genes or operons of SrsR were predicted based on adjacent genes and gene orientation (shown with bold words). Grey shading indicates genes that are not potential targets.

## Data Availability

The gSELEX data for SrsR were deposited in the ‘Transcription factor profiling of *Escherichia coli*’ (TEC) database at the National Institute of Genetics (Available online: https://shigen.nig.ac.jp/ecoli/tec/ (accessed on 1 April 2022)).

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
