# Peer review of "Transcription Factor SrsR (YgfI) Is a Novel Regulator for the Stress-Response Genes in Stationary Phase in Escherichia coli K-12"

_ijms, 2022, doi:10.3390/ijms23116055_

Round 1

Reviewer 1 Report

The research article entitled, “Transcription factor SrsR (YgfI) is a novel regulator for the stress response genes in stationary phase in Escherichia coli K-12" by Kobayashi et al., aims to explore the functional characterization of YgfI gene, an uncharacterized LysR family transcription factor in Escherichia coli.  

Authors conducted a genomic SELEX (gSELEX) screening to find regulation targets of YgfI and regulatory and phenotypic analyses were also performed. They found ten loci YgfI target genes on the E. coli genome and these loci were involved in some important stationary phase cellular functions such as biofilm formation, hydrogen peroxide resistance, and antibiotic resistance. These results suggested the participation of YgfI in biofilm formation and an increase in the tolerability against hydrogen peroxide hence authors have proposed to rename ygfI as srsR (a stress response regulator in stationary phase). While the topic is of increasing relevance, still, this reviewer has certain suggestions that would help produce a more comprehensive overview of the topic:

Comments:

  1. The English of manuscript can be polished and there are few typo errors in the manuscript that can be checked.
  2. Authors can explain that what induces expression of YgfI gene in E. coli?
  3. It would be beneficial/great interest to the readership if authors could summarize their findings in an illustrative figure.

4, Authors can add one paragraph for abbreviations.

5, Did authors check experimentally any antibiotic resistance characteristic of ygfI?

6, Authors should use p value in their figures instead of symbols such as *.

Author Response

Response to Reviewers

First of all, we appreciate both reviewers for the careful reading our manuscript and the useful comments for improvement of the manuscript. The major changes made in the revised version are:

  1. Additional Northern blot and RT-qPCR analysis experiments were performed in exponential phase cells to confirm transcriptional regulation by SrsR of the target genes involved in hydrogen peroxide tolerance under the same conditions as the hydrogen peroxide tolerance assay. These results are presented as Figure 7b and 7c, and the corresponding descriptions were added in the revised manuscript.
  2. The function of SrsR was reconsidered, and its predicted regulatory role within the host was described in the discussion section.

Response to Reviewer-1

The research article entitled, “Transcription factor SrsR (YgfI) is a novel regulator for the stress response genes in stationary phase in Escherichia coli K-12" by Kobayashi et al., aims to explore the functional characterization of YgfI gene, an uncharacterized LysR family transcription factor in Escherichia coli.

Authors conducted a genomic SELEX (gSELEX) screening to find regulation targets of YgfI and regulatory and phenotypic analyses were also performed. They found ten loci YgfI target genes on the E. coli genome and these loci were involved in some important stationary phase cellular functions such as biofilm formation, hydrogen peroxide resistance, and antibiotic resistance. These results suggested the participation of YgfI in biofilm formation and an increase in the tolerability against hydrogen peroxide hence authors have proposed to rename ygfI as srsR (a stress response regulator in stationary phase). While the topic is of increasing relevance, still, this reviewer has certain suggestions that would help produce a more comprehensive overview of the topic:

>>> We thank the reviewer for the positive feedback, and the valuable comments were used to improve our manuscript.

Comments:

1.The English of manuscript can be polished and there are few typo errors in the manuscript that can be checked.

>>> Thanks for this notice. Several spelling and italicization errors in gene names have been corrected.

2.Authors can explain that what induces expression of YgfI gene in E. coli?

>>> Since the expression of srsR is known to be repressed by Nac, a transcription factor expressed under nitrogen starvation conditions, we have added this statement into the DISCUSSION and the related citations were added in the REFERENCES.

3.It would be beneficial/great interest to the readership if authors could summarize their findings in an illustrative figure.

>>> We appreciate this suggestion. The story of regulatory function of SrsR is, however, not so complex, we decided not to add an illustrative figure.

4.Authors can add one paragraph for abbreviations.

>>> Abbreviations were added in the revised manuscript.

5.Did authors check experimentally any antibiotic resistance characteristic of ygfI?

>>> In this paper, we observed the two phenotype analyses of SrsR, biofilm formation and hydrogen peroxide resistance. The predicted SrsR-regulated genes such as YafQ (toxin of YafQ-DinJ tosin-antitoxin system) and YfdO (a prophage-coded protein) are known to involved in drug resistance. This finding is confirmed and described in the TEXT. 

6.Authors should use p value in their figures instead of symbols such as *.

>>> Thanks for this notice. Notation of p value has been corrected for Figures 6 and 7 in revised manuscript.

Reviewer 2 Report

The study by Kobayashi et ., investigated the function of the previously uncharacterized LysR family transcription factor (TF) SrsR (YgfI) by screening for its regulation targets/binding sites in E. coli K-12 using gSELEX. The authors have already successfully applied this approach in the characterization of various other TFs in previous works published in peer-reviewed journals. The binding of SrsR to the predicted by gSELEX regulatory targets was further experimentally verified in vitro in gel shift assays and the SrsR binding region upstream of yfdP was narrowed down in DNAse-I footprint analysis. Moreover, deletion/overexpression analysis suggested a role of SrsR in biofilm formation and resistance to H2O2. Even though this works could constitute a valuable contribution to characterization of novel TFs and expand our knowledge on gene regulation in E. coil K-12, there are several important points, which have to be addressed by the authors before the article can be considered for a publication.

The authors state that, based on their RT-PCR results, SrsR activates pgaABCD (line 204-205, data missing), which is involved in biofilm formation. Indeed, pgaA mutant has been shown to have reduced biofilm formation (Wang et al., 2004, J Bacteriol). This would actually mean that in the DsrsR no SrsR-dependent activation of pgaA can take place and this should result in less biofilm production than in WT. On the contrary, in the SrsR overexpression strain, there should be higher pgaA expression level than in WT and thus higher biofilm formation. The authors, however, observe the opposite effect of srsR deletion/SrsR overexpression in their biofilm assay (Fig 6). Please elaborate.

Figure 5 seems incomplete. RT-qPCR data for pgaA/D, ycgE/Z, ymgC, etc. are missing in the graph even though mentioned/referred to in the text (line 204-205, line 215-218) and primers are provided in the supplementary file. Please provide the missing data.

It is not clear how the H2O2 assay was performed with the SrsR overexpression strain: information missing in the material and method section. At what time point was SrsR overexpressed with IPTG and when was it subjected to treatment with H2O2?

The authors observed a stronger resistance to H2O2 of the strain carrying the SrsR overexpression vector (Fig. 7a), however did not analyze the mRNA levels of srsR, yfdO and yfdK in this background using norther blot (Fig. 7b). Please provide these data.

There seems to be major discrepancies between the cropped northern blot membranes presented in Figure 7b and the provided original membrane images: srsR probe – the background signal above the srsR band in WT appears much darker in Fig 7b than in the original image; yfdO probe – there seems to be a double band in WT in the original image, which is missing in Fig 7b; yfdK probe - there is no sharp band in WT and DsrsR in the original image, whereas there is one in Figure 7b. Please cross check and provide an explanation for the listed above differences.

Line 260-262: Sentence is confusing: “the srsR mRNA level in the srsR mutant strain was lower than that of the yfdONMLK transcript operon.” Please rewrite or explain meaning.

Line 248-250: “ mRNA levels of the wild type strain and srsR mutant strain in the exponential phase were determined using Northern blot analysis”; Line 269-270: “grown in LB medium at 37C for 6 hours in the exponential phase” (Figure 7b) and Line 398: “Total RNAs was extracted from stationary phase E. coli cells (OD600=1.5)”. Exponential or stationary phase? Please correct.

It is not clear if the p values obtained from the Student’s t-test in the statistical analysis were corrected for multiple testing (Fig. 6, 7A). If not, please correct for multiple testing for example using the Holm method.

The discussion section reads like an introduction/review of previously published data. Please rewrite.

Author Response

Response to Reviewers

First of all, we appreciate both reviewers for the careful reading our manuscript and the useful comments for improvement of the manuscript. The major changes made in the revised version are:

  1. Additional Northern blot and RT-qPCR analysis experiments were performed in exponential phase cells to confirm transcriptional regulation by SrsR of the target genes involved in hydrogen peroxide tolerance under the same conditions as the hydrogen peroxide tolerance assay. These results are presented as Figure 7b and 7c, and the corresponding descriptions were added in the revised manuscript.
  2. The function of SrsR was reconsidered, and its predicted regulatory role within the host was described in the discussion section.

Response to Reviewer-2

The study by Kobayashi et . investigated the function of the previously uncharacterized LysR family transcription factor YgfI (renamed to SrsR in this report) by screening for its regulation targets/binding sites in E. coli K-12 using gSELEX. The authors have already successfully applied this approach in the characterization of various other TFs in previous works published in peer-reviewed journals. The binding of SrsR to the predicted by gSELEX regulatory targets was further experimentally verified in vitro in gel shift assays and the SrsR binding region upstream of yfdP was narrowed down in DNAse-I footprint analysis. Moreover, deletion/overexpression analysis suggested a role of SrsR in biofilm formation and resistance to H2O2. Even though this works could constitute a valuable contribution to characterization of novel TFs and expand our knowledge on gene regulation in E. coil K-12, there are several important points, which have to be addressed by the authors before the article can be considered for a publication.

>>> We thank the reviewer for the valuable and thoughtful feedbacks to improve the manuscript. In response to the reviewers’ comments, we carefully rechecked the entire manuscript. In addition, we performed additional experiments, including Northern blotting assay and RT-qPCR assay. These results are shown in the revised Figure 7.  

The authors state that, based on their RT-PCR results, SrsR activates pgaABCD (line 204-205, data missing), which is involved in biofilm formation. Indeed, pgaA mutant has been shown to have reduced biofilm formation (Wang et al., 2004, J Bacteriol). This would actually mean that in the DsrsR no SrsR-dependent activation of pgaA can take place and this should result in less biofilm production than in WT. On the contrary, in the SrsR overexpression strain, there should be higher pgaA expression level than in WT and thus higher biofilm formation. The authors, however, observe the opposite effect of srsR deletion/SrsR overexpression in their biofilm assay (Fig 6). Please elaborate.

Figure 5 seems incomplete. RT-qPCR data for pgaA/D, ycgE/Z, ymgC, etc. are missing in the graph even though mentioned/referred to in the text (line 204-205, line 215-218) and primers are provided in the supplementary file. Please provide the missing data.

>>> Thanks for pointing these out. In the original manuscript, we included some information of SrsR functions obtained from the preliminary results using different experimental conditions. It also included results of not only the direct regulatory targets of SrsR but also the indirect targets. In order to avoid confusions, we have included in the revised version only the direct targets of SrsR. Accordingly, in the revised manuscript, the relevant descriptions have been modified by removing unnecessary data from TEXT and Table S1 (Primers list).

Direct target genes of SrsR such as bluR and cdgI have been reported to play an inhibitory role of biofilm formation. In fact, we found that these genes are activated by SrsR under our conditions employed in this study, and the activated target proteins repressed biofilm formation as detected by the biofilm assay. Then, we concluded that SrsR plays inhibitory role of biofilm formation.

It is not clear how the H2O2 assay was performed with the SrsR overexpression strain: information missing in the material and method section. At what time point was SrsR overexpressed with IPTG and when was it subjected to treatment with H2O2?

>>> Thanks for this notice. When observing the effect of SrsR overexpression in the H2O2 assay throughout this study, we detailed assay conditions in 'Bacterial strains and plasmids' in Materials and Methods section that IPTG at a final concentration of 0.5 mM was added from the start of inoculation. It was also specified that the hydrogen peroxide resistance was observed when the growth of E. coli reached to OD600=0.4.

The authors observed a stronger resistance to H2O2 of the strain carrying the SrsR overexpression vector (Fig. 7a), however did not analyze the mRNA levels of srsR, yfdO and yfdK in this background using norther blot (Fig. 7b). Please provide these data.

There seems to be major discrepancies between the cropped northern blot membranes presented in Figure 7b and the provided original membrane images: srsR probe – the background signal above the srsR band in WT appears much darker in Fig 7b than in the original image; yfdO probe – there seems to be a double band in WT in the original image, which is missing in Fig 7b; yfdK probe - there is no sharp band in WT and DsrsR in the original image, whereas there is one in Figure 7b. Please cross check and provide an explanation for the listed above differences.

>>> To reevaluate the effect of SrsR on the expression of target genes under the test conditions of H2O2 resistance (in exponential phase, OD600=0.4), we performed additional experiments of Northern blotting and RT-qPCR analysis (Figure 7b and 7c in the revised manuscript). Northern blotting showed the increased level of yfdONMLK mRNA by SrsR overexpression. Since the transcript of yfdONMLK operon was detected as a single band, we believe our new data are more plausible than those presented in the original manuscript (see original image file). However, little effect was observed for the srsR deletion strain supposedly due to the low-level expression of yfdONMLK in the absence of SrsR. To confirm this prediction, RT-qPCR analysis was performed, and the result indicated that the transcript level was indeed reduced in the srsR-deletion strain compared to the wild-type strain. Based on these results, the relevant sections of the text have been revised.

Line 260-262: Sentence is confusing: “the srsR mRNA level in the srsR mutant strain was lower than that of the yfdONMLK transcript operon.” Please rewrite or explain meaning.

>>> We agree this point. The relevant sentence has been corrected in revised text.

Line 248-250: “ mRNA levels of the wild type strain and srsR mutant strain in the exponential phase were determined using Northern blot analysis”; Line 269-270: “grown in LB medium at 37C for 6 hours in the exponential phase” (Figure 7b) and Line 398: “Total RNAs was extracted from stationary phase E. coli cells (OD600=1.5)”. Exponential or stationary phase? Please correct.

>>> Thanks for this notice. Total RNAs for Northern blot analysis was extracted from exponential phase, and the text was corrected.

It is not clear if the p values obtained from the Student’s t-test in the statistical analysis were corrected for multiple testing (Fig. 6, 7A). If not, please correct for multiple testing for example using the Holm method.

>>> Notation of p value has been corrected for Figures 6 and 7 in revised manuscript.

The discussion section reads like an introduction/review of previously published data. Please rewrite.

>>> The DISCUSSION was simplified by removing the redundant portions including the references to transcription factors associated with oxidative stress. Instead, the physiological role of SrsR in the oxidative stress tolerance was added.

Round 2

Reviewer 2 Report

The revised manuscript of Kobayashi et al. has been substantially improved by the addition of new experiments and re-writing sections of the paper. There is only minor criticism left from my side:

The original image of the new northern blot is missing. The authors could provide it as a supplementary figure.

It is still not included in the text if the given p values obtained from the t-tests were corrected for multiple comparison. This should be clearly stated.

Could the authors include an extra paragraph in the discussion elaborating on the biological importance and interconnection of the two roles of SrsR: on one hand side suppressing biofilm formation and on the other increasing resistance to oxygen peroxide and mention if other regulators with similar dual effects are known.

Author Response

Response to Reviewers

First of all, we appreciate the reviewer for the careful reading our manuscript and the useful comments for improvement of the manuscript.

Response to Reviewer-2

The revised manuscript of Kobayashi et al. has been substantially improved by the addition of new experiments and re-writing sections of the paper. There is only minor criticism left from my side:

The original image of the new northern blot is missing. The authors could provide it as a supplementary figure.

>>> We attached the file with track changes. The images of the membrane was added at the end of the file.

It is still not included in the text if the given p values obtained from the t-tests were corrected for multiple comparison. This should be clearly stated.

>>> Colored text has been added to the relevant text below in the revised text.

Statistically significant differences were obtained using the Student’s t-test with multiple comparisons correction and are represented by line over bars with associated P values.

Could the authors include an extra paragraph in the discussion elaborating on the biological importance and interconnection of the two roles of SrsR: on one hand side suppressing biofilm formation and on the other increasing resistance to oxygen peroxide and mention if other regulators with similar dual effects are known.

>>> We thank the reviewer for raising these discussion points. After the consideration, we have added the extra paragraph to discussion in the revised text.

While SrsR suppressed biofilm formation, SrsR increased the tolerability against hydrogen peroxide (Figure 7). E. coli carries several regulators for activating a set of genes for oxidative stress, including OxyR, SoxR, and SoxS [37]. Recently, it was reported that Cys25 of OxyR, which is S-nitrosylation in planktonic cells, plays an important role in suppressing biofilm formation [38]. Such transcriptional regulation by SrsR and OxyR suggests the importance of switching between the biofilm and planktonic lifestyles as well as conferring oxidative stress tolerance to planktonic cells in bacterial adaptation to the stressful environment.
